# Answering Complex Causal Queries With the Maximum Causal Set Effect (MCSE)

**Zachary Markovich**[*]
Massachussetts Institute of Technology
Cambridge, MA, 02139
zmarko@mit.edu

## Abstract

The standard tools of causal inference have been developed to answer simple causal queries which can be easily formalized as a small number of statistical estimands in the context of a particular structural causal model (SCM); however, scientific theories often make diffuse predictions about a large number of causal variables. This article proposes a framework for parameterizing such complex causal queries as the maximum difference in causal effects associated with two sets of causal variables that have a researcher specified probability of occurring. We term this estimand the *Maximum Causal Set Effect* (MCSE) and develop an estimator for it that is asymptotically consistent and conservative in finite samples under assumptions that are standard in the causal inference literature. This estimator is also asymptotically normal and amenable to the non-parametric bootstrap, facilitating classical statistical inference about this novel estimand. We compare this estimator to more common latent variable approaches and find that it can uncover larger causal effects in both real world and simulated data.

## 1 Introduction

Recent advances in machine learning technology have made it possible to non-parametrically estimate many parameters present in complex structural causal models (SCMs). Specifically, such estimating technology has rapidly advanced for three major causal inference settings: the many causes setting, the many moderators setting, and the many mediators setting. All three settings represent a situation in which a particular causal query can be stated in terms of a large number of combinations of different variables. Specifically, a researcher could estimate a different treatment effect associated with each of the many different possible combinations of causes [Imbens, 2000, Wang and Blei, 2019, Li et al., 2019, Wang et al., 2018, Zheng et al., Forthcoming], a different conditional treatment effect for each of the many different combinations of moderators [Green and Kern, 2012, Athey and Imbens, 2016, Grimmer et al., 2017, Wager and Athey, 2018, Künzel et al., 2019], and a different mediated effect for each of the many different combinations of mediators [Zhou and Yamamoto, 2020, Daniel et al., 2015]. Such causal queries are complex in the sense that they require summarizing the combined influence of a large number of causal variables.

The main challenge for applied researchers in such settings is that standard causal inference algorithms are designed to provide a different estimate associated with each of the many causal variables rather than a single number summarizing the combined influence of all the causal variables together. Consider, for example, the setting of inferring the causal effect of actors on a film's box office performance. Wang and Blei [2019] provide a framework for estimating the average treatment effect associated with every actor on a film's performance. While certainly useful for making predictions

---

[*]PHD Candidate, Interdisciplinary Doctoral Program in Statistics and Political Science, https://zmarkovich.github.io/

35th Conference on Neural Information Processing Systems (NeurIPS 2021).

about which actors a director should cast, an economist studying the film industry might prefer a single number which summarizes the general importance of actors in general for a film's box office success. As discussed in the next section, such settings are common in scientific research, suggesting the need for novel causal estimands to parameterize the predictions of such theories in the context of a particular SCM.

**Contribution**    The contribution of this paper is threefold. First, it introduces the notion of a complex causal query and argues that existing causal estimands are of limited utility to applied researchers in the face of such queries. Second, it defines a novel estimand – the *Maximum Causal Set Effect* (MCSE) – which can be used to provide an interpretable answer to such complex queries. Finally, the paper introduces an estimator for this estimand. The estimator is based on techniques proposed in the double Q-learning literature [Hasselt, 2010] and is asymptotically consistent and conservative in finite samples under assumptions that are standard in the causal inference literature. It is also asymptotically normal and amenable to non-parametric bootstrap techniques, facilitating classical statistical inference about the MCSE.

## 2    Setting and Previous Work

### 2.1    Problem Overview

Standard approaches to causal inference [Pearl, 2009] typically begin with the researcher specifying an SCM and then defining a causal query which can be answered based on the assumed SCM. Under certain assumptions about the SCM, it may be possible to estimate the answer to that causal query using the conventional tools of statistical inference. The standard tools of causal inference are designed with settings in mind where the predictions of a scientific theory take the form of a *simple causal query*. Such queries are stated in terms of some low dimensional *causal variable* $\mathbf{t}$ and some outcome $Y$. For example, a question like how much does a medical procedure reduce the risk of disease, represents a simple causal query because it is defined in terms of a single unidimensional treatment. Such queries can be easily quantified using conventional statistical estimands because they are directly formulated in terms of a small number of theoretically motivated variables.

This paper instead focuses on situations where a scientific theory makes diffuse predictions about the importance of a large number of causal variables, defying the stylization of simple causal queries. Such queries are common in scientific research. For example:

- **Genome Wide Association Studies (GWAS)** – GWAS attempt to quantify the causal effect of a huge number of individual genotypes on the likelihood that some trait is expressed [Stephens and Balding, 2009, Visscher et al., 2017].

- **Personality** – psychologists are often interested in the effect certain personality traits (such as extraversion or neuroticism) might have on life outcomes [Pervin, 2003], but such traits are only observed by the researcher as responses to a large number of survey questions.

- **Text** – language is complex and multi-faceted and the causal effect of the wording of a document on a user's response requires an assessment of the contribution of many different topics or words together [Fong and Grimmer, 2016, Egami et al., 2018, Fong and Grimmer, forthcoming].

- **Complex medical treatments** – many medical treatments cannot be reduced to a single low dimensional representation. For example, radiation exposure is observed as a high dimensional vector [Nabi et al., 2017] and medical researchers might also wish to understand the combined importance of many procedures using electronic medical records [Gottesman et al., 2013].

Such causal queries are *complex* because they require estimating the joint influence of many causal variables.

The SCM undergirding such complex queries can take many forms. Three major examples are: (a) the many causes setting where the researcher wishes to understand the joint influence of many treatments (b) many moderators setting where the researcher wishes to understand how effect of a binary treatment varies based on many variables (c) the many mediators setting where the researcher wishes to model how a causal effect can be decomposed into many different pieces. These SCM's are

Figure 1: Visualization of Causal Graphs With Complex Queries

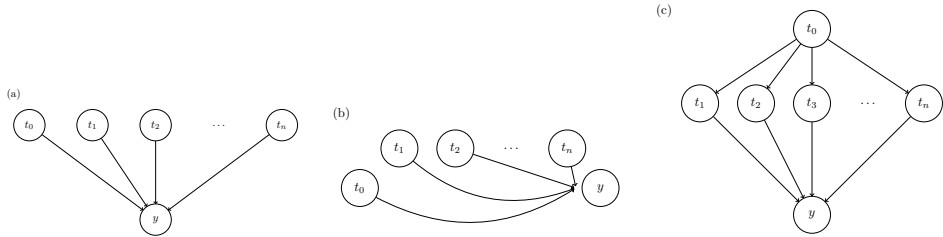

**Note**: Figure visualizes SCMs corresponding to complex causal queries in the case of (a) many causes, (b) many moderators, and (c) many mediators. (a) visualizes the case where treatment types $t_0 \dots t_n$ each influence $y$ described in Wang and Blei [2019]. (b) visualizes the case where the causal effect of $t_0$ on $y$ is directly modified by $t_1 \dots t_n$ as described in VanderWeele and Robins [2007]. (c) visualizes the case where the effect of $t_0$ on $y$ is mediated by $t_1 \dots t_n$ as described by Zhou and Yamamoto [2020].

visualized in Figure 1 in the form of directed acylical graphs (DAGs). The unifying trait of a complex causal query is that it asks about the importance of many arrows present in each DAG.

Techniques developed in the context of simple causal queries cannot be readily used to answer complex ones. While the standard tools of causal inference can be used to estimate causal effects corresponding to every combination of causal variables in SCM's like those visualized in Figure 1, they do not provide applied researchers with a single unambiguous estimate with which to summarize the joint causal effect of many such variables.

## 2.2 Previous Work

The only existent proposal for addressing the challenge presented by complex causal queries in the machine learning literature is to dimension reduce the relevant causal variables and then focus on a simple causal query defined in terms of that latent trait [Fong and Grimmer, 2016, forthcoming, Nabi et al., 2017]. This strategy has only been proposed in the many causes setting, but could also be extended to the many moderators or many mediators cases as well. Such a strategy is inherently reductive and risks understating the magnitude of causal effects because it disregards all variation in the treatment types that is not accounted for in the latent trait. Additionally such latent traits are often scale invariant and so may lack a scientifically meaningful interpretation.

## 2.3 Assumptions and Notation

We assume that the researcher observes a set of $N$ independent $(\mathbf{t}_i, Y_i, \mathbf{x}_i)$ triplets where $Y_i$ is the outcome, and $\mathbf{t}_i$ is a length $K$ vector indicating the *treatment type* received by unit $i$, and $\mathbf{x}_i$ is a length $J$ vector representing a set of background covariates that causal effects should be adjusted for. Additionally, let $\mathcal{T}$ denote the support of the distribution of $\mathbf{t}_i$.

We also assume that the researcher has knowledge of the population distribution of $\mathbf{t}_i$: $g(\mathbf{t})$. In many settings, the empirical distribution of $\mathbf{t}_i$ will be the most logical choice, but other choices may be reasonable as well if the population distribution is known to the researcher, as might be the case when conducting survey research or if the treatment types were experimentally randomized.

Finally, we assume that the researcher has specified some SCM and has specified a simple causal query, $\tau(\mathcal{T}', \mathcal{T}'')$, which is defined in terms of two subsets: $\mathcal{T}', \mathcal{T}'' \subseteq \mathcal{T}$. In the many causes case, $\tau(\mathcal{T}', \mathcal{T}'')$ might take the form:

$$\tau(\mathcal{T}', \mathcal{T}'') \equiv \mathbb{E}\left(\mathbb{E}\left(Y_i | \mathrm{do}(t)\right) | t \in \mathcal{T}'\right) - \mathbb{E}\left(\mathbb{E}\left(Y_i | \mathrm{do}(t)\right) | t \in \mathcal{T}''\right)$$

where $\mathrm{do}(\cdot)$ represents some causal intervention [Pearl, 2009]. This estimand represents the average effect of receiving a set of treatments contained in $\mathcal{T}'$ rather than $\mathcal{T}''$.

In the many moderators case on the other hand let the zeroth element of the treatment types vector recieved by unit $i$, $\mathbf{t}_{i,0} \in \{0, 1\}$, denote the level of some binary treatment received by unit $i$. Similarly, let the remaining elements of $\mathbf{t}_i$ be denoted $\mathbf{t}_{i,-0}$ and indicate the level received by unit $i$ on the many moderators. Then a possible choice for $\tau(\mathcal{T}', \mathcal{T}'')$ might be:

$$\tau(\mathcal{T}', \mathcal{T}'') \equiv \mathbb{E}\left(\mathbb{E}\left(Y_i | \mathrm{do}(\mathbf{t}_{i,0} = 1)\right) - \mathbb{E}\left(Y_i | \mathrm{do}(\mathbf{t}_{i,0} = 0)\right) | \mathbf{t}_{i,-0} \in \mathcal{T}'\right)$$
$$- \mathbb{E}\left(\mathbb{E}\left(Y_i | \mathrm{do}(\mathbf{t}_{i,0} = 1)\right) - \mathbb{E}\left(Y_i | \mathrm{do}(\mathbf{t}_{i,0} = 0)\right) | \mathbf{t}_{i,-0} \in \mathcal{T}''\right)$$

which represents the difference in average treatment effects between units with moderators contained in $\mathcal{T}'$ rather than $\mathcal{T}''$. Defining such estimands in the many mediators case requires more cumbersome notation, but can be accomplished in terms of an average of different combinations of path specific effects [Daniel et al., 2015, Zhou and Yamamoto, 2020].

# 3 The Maximum Causal Set Effect

The challenge for applied researchers in the presence of such complex causal queries is that a different value of $\tau(\mathcal{T}', \mathcal{T}'')$ can be defined for every distinct pair of sets $\mathcal{T}', \mathcal{T}'' \subseteq \mathcal{T}$, leaving the analyst without a single unambiguous causal estimand to summarize their findings. In this section, we define a causal quantity of interest which overcomes this challenge by focusing on the contrast between two sets $\mathcal{T}_q^{\mathrm{Max}}$ and $\mathcal{T}_q^{\mathrm{Min}}$ which maximize $\tau(\mathcal{T}', \mathcal{T}'')$. To avoid choosing sets $\mathcal{T}_q^{\mathrm{Max}}$ and $\mathcal{T}_q^{\mathrm{Min}}$ which correspond to unrepresentative edge cases, we require that the sets be of a researcher specified size: $q$. Formally, let the set of subsets of $\mathcal{T}$ such that the probability that $\mathbf{t}_i$ is in $\mathcal{T}$ is at least $q$ be defined as: $\mathcal{T}_q \equiv \{\mathcal{T}' \subseteq \mathcal{T} : P(\mathbf{t}_i \in \mathcal{T}') \geq q\}$ where $P(\mathbf{t}_i \in \mathcal{T}') = \int_{\mathcal{T}} g(\mathbf{t}) \mathbb{1}\{\mathbf{t} \in \mathcal{T}\} d\mathbf{t}$.

We then define $\mathrm{MCSE}_q$ as:

$$\mathrm{MCSE}_q = \max_{\mathcal{T}', \mathcal{T}'' \in \mathcal{T}_q} \tau(\mathcal{T}', \mathcal{T}'') = \tau\left(\mathcal{T}_q^{\mathrm{Max}}, \mathcal{T}_q^{\mathrm{Min}}\right)$$

We refer to $\mathcal{T}_q^{\mathrm{Max}}$ as the *maximum causal set* and $\mathcal{T}_q^{\mathrm{Min}}$ as the *minimum causal set*. For many applications, the MCSE will have an intuitive and scientifically meaningful interpretation. In the actors example, it might be used to answer a question like what is the expected difference in box office performance between a film cast with one of the 10% best performing casts rather than one of the bottom 10% worst performing casts? Similarly, in the genetics example, it might answer the question, what is the difference in the efficacy of some drug for patients with one of the top 10% most treatment enhancing sets of genes rather than one of the bottom 10% most treatment diminishing sets of genes?

# 4 Estimation

This section outlines an algorithm for estimating $\mathrm{MCSE}_q$. Sample splitting is a major part of this algorithm and this section develops the procedure in the context of a single data split. The efficiency of this estimator can also easily be improved by rotating the roles that each subset of the data plays and then averaging the results, a procedure known as crossfitting [Chernozhukov et al., 2017], which we discuss in Appendix A.

## 4.1 Algorithm Overview

A basic result in the Q-learning literature is that a single sample estimator for the maximum expected value will have an upward bias [Hasselt, 2010]. Since conservative estimators are easier to interpret and necessary for valid hypothesis testing, we follow the lead of Hasselt [2010] in using a split sample estimator for this estimation task. This approach is also useful in demonstrating the asymptotic normality of the resulting estimator as well.

Specifically, we begin by assuming that the analyst has randomly split the observations into two equally sized sets, $\mathcal{S}^{\mathrm{Est}}$ and $\mathcal{S}^{\mathrm{Prob}}$. We further assume that the analyst has specified two models. The first uses the elements of the splitting set to make predictions about the probability that any $\mathcal{T}', \mathcal{T}'' \in \mathcal{T}_q$ are the true maximum and minimum causal sets and we denote its predictions: $\hat{P}(\mathcal{T}' =$

$\mathcal{T}_q^{\text{Max}} \cap \mathcal{T}'' = \mathcal{T}_q^{\text{Min}}$). The second model makes a prediction about $\tau(\mathcal{T}', \mathcal{T}'')$ for any two $\mathcal{T}', \mathcal{T}''' \subseteq \mathcal{T}$, and we denote its predictions $\hat{\tau}(\mathcal{T}', \mathcal{T}'')$. Note $\hat{P}(\mathcal{T}' = \mathcal{T}_q^{\text{Max}} \cap \mathcal{T}'' = \mathcal{T}_q^{\text{Min}})$ should make use only of outcomes that are included in $\mathcal{S}^{\text{Prob}}$ while $\hat{\tau}(\mathcal{T}', \mathcal{T}'')$ should only use the outcomes in $\mathcal{S}^{\text{Est}}$ so that, $\forall \mathcal{T}', \mathcal{T}'' \in \mathcal{T}_q$, $\hat{P}(\mathcal{T}' = \mathcal{T}_q^{\text{Max}} \cap \mathcal{T}'' = \mathcal{T}_q^{\text{Min}}) \perp\!\!\!\perp \hat{\tau}(\mathcal{T}', \mathcal{T}'')$ conditional on observing the sample values of $\mathbf{t}_i$ and $x_i$ for all units. After specifying models for $\hat{P}(\mathcal{T}' = \mathcal{T}_q^{\text{Max}} \cap \mathcal{T}'' = \mathcal{T}_q^{\text{Min}})$ and $\hat{\tau}(\mathcal{T}', \mathcal{T}'')$, estimation proceeds as a weighted average of the estimates for $\hat{\tau}(\mathcal{T}', \mathcal{T}'')$ for every $\mathcal{T}', \mathcal{T}'' \in \mathcal{T}_q$:

$$\widehat{\text{MCSE}}_q = \sum_{\mathcal{T}', \mathcal{T}'' \in \mathcal{T}_q} \hat{P}(\mathcal{T}' = \mathcal{T}_q^{\text{Max}} \cap \mathcal{T}'' = \mathcal{T}_q^{\text{Min}}) \hat{\tau}(\mathcal{T}', \mathcal{T}'')$$

## 4.2 Point Estimation Properties

A major requirement for the good behavior of this estimator is that $\hat{P}(\mathcal{T}' = \mathcal{T}_q^{\text{Max}} \cap \mathcal{T}'' = \mathcal{T}_q^{\text{Min}})$ obey the basic probability axioms and that $\hat{P}(\mathcal{T}' = \mathcal{T}_q^{\text{Max}} \cap \mathcal{T}'' = \mathcal{T}_q^{\text{Min}})$ assign zero probability to subsets of $\mathcal{T}$ not in $\mathcal{T}_q$. These requirements are entirely verifiable by the analyst through the careful construction of $\hat{P}(\mathcal{T}' = \mathcal{T}_q^{\text{Max}} \cap \mathcal{T}'' = \mathcal{T}_q^{\text{Min}})$ and are formalized in the following assumption:

**Assumption 1.** $\hat{P}(\mathcal{T}' = \mathcal{T}_q^{Max} \cap \mathcal{T}'' = \mathcal{T}_q^{Min})$ *satisfies the following conditions:*

- $\sum_{\mathcal{T}', \mathcal{T}'' \in \mathcal{T}_q} \hat{P}(\mathcal{T}' = \mathcal{T}_q^{Max} \cap \mathcal{T}'' = \mathcal{T}_q^{Min}) = 1$

- $\forall \mathcal{T}', \mathcal{T}'' \in \mathcal{T}_q, 0 \leq \hat{P}(\mathcal{T}' = \mathcal{T}_q^{Max} \cap \mathcal{T}'' = \mathcal{T}_q^{Min}) \leq 1$

- $\hat{P}(\mathcal{T}' = \mathcal{T}_q^{Max} \cap \mathcal{T}'' = \mathcal{T}_q^{Min}) = 0$ *for all* $\mathcal{T}', \mathcal{T}'' \notin \mathcal{T}_q$

Under Assumption 1, $\widehat{\text{MCSE}}_q$ can be interpreted as a weighted average of estimators for the causal effect of being treated with a treatment type in one set rather than another. Because $\text{MCSE}_q$ is defined as the maximum of such causal effects for any two subsets of $\mathcal{T}$ of the required size, it will always be greater than the expectation of this average, leading to the following proposition:

**Proposition 1.** *If* $\forall \mathcal{T}', \mathcal{T}'' \in \mathcal{T}_q$, $\mathbb{E}\left(\hat{\tau}(\mathcal{T}', \mathcal{T}'')\right) \leq \tau(\mathcal{T}', \mathcal{T}'')$ *and the conditions of Assumption 1 hold, then:*

$$\mathbb{E}\left(\widehat{\text{MCSE}}_q\right) \leq \text{MCSE}_q$$

Proof in appendix C.1

The conditions for finite sample conservatism are relatively mild (for example, $\hat{P}(\mathcal{T}' = \mathcal{T}_q^{\text{Max}} \cap \mathcal{T}'' = \mathcal{T}_q^{\text{Min}})$ could be misspecified or inconsistent); however, as formalized in the next proposition, the conditions for the consistency of $\text{MCSE}_q$ are a bit stronger and require that $\hat{P}(\mathcal{T}' = \mathcal{T}_q^{\text{Max}} \cap \mathcal{T}'' = \mathcal{T}_q^{\text{Min}})$ converge to a binary indicator identifying $\mathcal{T}_q^{\text{Min}}$ and $\mathcal{T}_q^{\text{Max}}$:

**Proposition 2.** *If* $\forall \mathcal{T}', \mathcal{T}'' \in \mathcal{T}_q$,

$$\hat{P}(\mathcal{T}' = \mathcal{T}_q^{Max} \cap \mathcal{T}'' = \mathcal{T}_q^{Min} | \mathcal{S}^{Prob}) \xrightarrow[n \to \infty]{p} \mathbb{1}\{\mathcal{T}' = \mathcal{T}_q^{Max}\} \mathbb{1}\{\mathcal{T}' = \mathcal{T}_q^{Max}\}$$

*and*

$$\hat{\tau}(\mathcal{T}', \mathcal{T}'') \xrightarrow[n \to \infty]{p} \tau(\mathcal{T}', \mathcal{T}'')$$

*then*

$$\widehat{\text{MCSE}}_q \xrightarrow[n \to \infty]{p} \text{MCSE}_q$$

*This result will also hold if convergence in probability is replaced with almost sure convergence.*

Proof in Appendix C.2.

Some machine learning techniques (e.g. support vector machines, regression trees, etc.) will not readily produce probabilistic estimates for $\hat{P}(\mathcal{T}' = \mathcal{T}_q^{\text{Max}} \cap \mathcal{T}'' = \mathcal{T}_q^{\text{Min}})$, instead generating only a

binary prediction for the two sets $\mathcal{T}_q^{\text{Max}}$ and $\mathcal{T}_q^{\text{Min}}$.[2] The following proposition shows that such binary estimators will perform at best as well as probabilistic estimators as long as the two estimators have the same expectation:

**Proposition 3.** *Let, $d(\mathcal{T}', \mathcal{T}'') \in \{0, 1\}$ and $w(\mathcal{T}', \mathcal{T}'') \in [0, 1]$ represent two choices for $\hat{P}(\mathcal{T}' = \mathcal{T}_q^{Max} \cap \mathcal{T}'' = \mathcal{T}_q^{Min})$. Let $\widehat{MCSE}_q^{\,d}$ and $\widehat{MCSE}_q^{\,w}$ represent the corresponding estimators for $MCSE_q$. Then if $\forall \mathcal{T}', \mathcal{T}'' \in \mathcal{T}_q, \mathbb{E}\left(d(\mathcal{T}', \mathcal{T}'')\right) = \mathbb{E}\left(w(\mathcal{T}', \mathcal{T}'')\right),$*

$$\mathbb{E}\left(\left(MCSE_q - \widehat{MCSE}_q^{\,w}\right)^2\right) \leq \mathbb{E}\left(\left(MCSE_q - \widehat{MCSE}_q^{\,d}\right)^2\right)$$

Proof in Appendix C.3

A direct implication of this result is that bootstrap aggregation can be used to improve the performance of any binary predictor for $\hat{P}(\mathcal{T}' = \mathcal{T}_q^{\text{Max}} \cap \mathcal{T}'' = \mathcal{T}_q^{\text{Min}})$ to create a probabilistic estimator without changing the expected value of the predictions.

### 4.3 Interval Estimation

While the previous section establishes the properties of the point estimator for $\text{MCSE}_q$, such results will be of little utility for applied researchers without a corresponding framework for measuring the uncertainty of those estimates. In this section, we begin the process of providing such results by introducing the assumption that $\hat{\tau}(\mathcal{T}', \mathcal{T}'')$ can be represented as a linear combination of the estimation set outcomes:

**Assumption 2.** *Let $Z = \{\mathbf{t}_i, x_i : i \in \mathcal{S}^{Est}\}$. For any $\mathcal{T}', \mathcal{T}'' \in \mathcal{T}_q$ there exists a set of transformations $\{f_i(Z, \mathcal{T}', \mathcal{T}'') : i \in \mathcal{S}^{Est}\}$ such that:*

$$\hat{\tau}(\mathcal{T}', \mathcal{T}'') = \sum_{i \in \mathcal{S}^{Est}} f_i(Z, \mathcal{T}', \mathcal{T}'') Y_i$$

Many common estimators for causal effects (e.g. matching, weighting, regression techniques, etc) fit this form, so such an assumption will not be unduly restrictive in many settings.

This assumption eases the derivation of asymptotic normality because it shows that $\hat{\tau}(\mathcal{T}', \mathcal{T}'')$ can be represented as the sum of independent random variables. The following proposition uses the central limit theorem derived by Neumann [2013] to show that multiplication by $\hat{P}(\mathcal{T}' = \mathcal{T}_q^{\text{Max}} \cap \mathcal{T}'' = \mathcal{T}_q^{\text{Min}})$ will not impact this convergence so that asymptotic normality of $\widehat{\text{MCSE}}_q$ can be preserved under some mild regularity conditions:

**Proposition 4.** *If $\hat{P}(\mathcal{T}' = \mathcal{T}_q^{Max} \cap \mathcal{T}'' = \mathcal{T}_q^{Min})$ satisfies assumption 1; $\forall i, \mathbb{E}\left(Y_i^2\right) < \infty$; and $\forall \epsilon > 0,$*

$$\sum_{i \in \mathcal{S}^{Est}} \frac{1}{|\mathcal{S}^{Est}|} \mathbb{E}\left(f_i(Z, \mathcal{T}', \mathcal{T}'')^2 Y_i^2 \mathbb{1}\{|f_i(Z, \mathcal{T}', \mathcal{T}'')| > \epsilon\}\right) \xrightarrow[|\mathcal{S}^{Est}| \to \infty]{} 0$$

*Then, conditional on observing the estimation set values of $\mathbf{t}_i$ and $\mathbf{x}_i$,*

$$\frac{\left(\widehat{MCSE}_q - \mathbb{E}\left(\widehat{MCSE}_q\right)\right)}{\sqrt{Var(\widehat{MCSE}_q)}} \xrightarrow{D} \mathcal{N}(0, 1)$$

Proof in Appendix C.4

The final result necessary for conducting classical statistical inference is a corresponding variance estimator. This can be most easily accomplished via the non-parametric bootstrap. Specifically,

---

[2]Note, many of these algorithms can be tweaked to provide such probabilistic estimates. For example, Bayesian regression trees [Chipman et al., 2010] is a tree based method that can easily make these sorts of probabilistic predictions. Proposition 3 suggest that such approaches would also be preferable to a coarse binary prediction.

Mammen [1992] shows that the non-parametric bootstrap is consistent for an asymptotically normal estimator that can be represented as a linear transformation of some set of independent observations. The following lemma uses assumption 2 to provide just such a result:

**Lemma 1.**

$$\widehat{MCSE}_q = \sum_{i \in \mathcal{S}^{Est}} Y_i w_i$$

*where* $w_i = \sum_{\mathcal{T}', \mathcal{T}'' \in \mathcal{T}_q} \hat{P}(\mathcal{T}' = \mathcal{T}_q^{Max} \cap \mathcal{T}'' = \mathcal{T}_q^{Min}) f_i(Z, \mathcal{T}', \mathcal{T}'')$

*Proof.* The proof follows trivially by using assumption 2 to substitute $\sum_{i \in \mathcal{S}^{Est}} f_i(Z, \mathcal{T}', \mathcal{T}'')$ for $\hat{\tau}(\mathcal{T}', \mathcal{T}'')$ in the definition of $\widehat{MCSE}_q$ and then changing the order of summation. □

So the variance and confidence intervals of $\widehat{MCSE}_q$ can be consistently estimated by bootstrap resampling from the set $\{Y_i w_i : i \in \mathcal{S}^{Est}\}$.[3]

# 5 Experiments

## 5.1 Benchmarks on Synthetic Data

We first consider the performance of this estimation procedure using synthetic data. Specifically, to asses the performance of this estimator, we implemented it on a synthetic version of the many causes setting. First, we generated a set of $N$ length $K$ vectors of causes for each unit $i$ as $\mathbf{t}_i \sim \mathcal{N}(0, \Sigma)$ where $\Sigma$ is some matrix with ones on the diagonal elements and some value $\rho \in [0, 1]$ in the off diagonal elements. We then generated the outcome as $\mu_i = \mathbf{t}_i' \beta$ where $\beta$ is a length $K$ vector composed of i.i.d draws from the standard normal distribution. Finally, we normalized $\mu_i$ so that the corresponding value of $MCSE_q$ was always 1 and generated the outcome variables as $Y_i = \mu_i + \epsilon_i$ where $\epsilon_i \sim \mathcal{N}(0, 1)$.

We implemented two estimators on this dataset. The first is the split sample $\widehat{MCSE}_q$ estimator described in this paper[4]. Note that under this simulation set up, all the assumptions needed for the theoretical results presented in Section 4 to hold are known to be true, so $\widehat{MCSE}_q$ should be unbiased and consistent. We compared the performance of $\widehat{MCSE}_q$ with an estimate for $\widehat{MCSE}_q$ generated using a linear regression of $Y_i$ on the first principal component of $\mathbf{t}_i$.[5] This estimator corresponds to the current state of the art for drawing causal inferences in the face of a complex causal query, which involves using dimension reduction techniques to simplify the complex causal query into a simple one. We repeated this procedure 100 times for each combination of $K = 2$, 10, and 50; $\rho = 0$, .5 and, .9; and values of $N$ between 100 and 1,000.

Figure 2 visualizes the results of this analysis. Each point in the figure represents the average of all 300 iterations of the simulation procedure with the same values of $n$ and $K$ or $n$ and $\rho$.[6] Because the bias of both estimators is large relative to their variance in this setting, Figure 2 focuses on the bias of the estimators.[7] These estimates show that $\widehat{MCSE}_q$ is a large improvement over the latent trait model, generating significantly less biased estimates even when $\rho$ is large and the principal components analysis (PCA) should perform well. Importantly, the bias of $\widehat{MCSE}_q$ appears to vanish asymptotically while the PCA estimator shows little convergence as the sample size increases.[8]

---

[3]Note, clustered standard errors can also be easily generated using the block bootstrap.

[4]Specifically, one using monte carlo sampling from the asymptotic distribution of linear regression of $Y_i$ on $\mathbf{t}_i$ as $\hat{P}(\mathcal{T}' = \mathcal{T}_q^{Max} \cap \mathcal{T}'' = \mathcal{T}_q^{Min})$ and a linear regression for $\hat{\tau}(\mathcal{T}', \mathcal{T}'')$. See Appendix B.1 for more details on the implementation of $\widehat{MCSE}_q$

[5]See appendix B.2 for details on the implementation of this estimator.

[6]Note, the monte carlo error in these estimates is quite low. The standard error associated with these average is never higher than .019 for any of the points.

[7]Appendix 4 presents estimates for the root mean squared error, which show a similar pattern

[8]Note additional simulation results are also presented in Sections B.3 and B.4 of the supplementary information.

Figure 2: Simulation Results

**Note:** Red dots identify the bias of the method for quantifying the combined effect of many causes proposed in this paper while blue dots show the bias of dimension reduction techniques that represent the current state of the art for this same task.

## 5.2 An Application to Real World Data

Our second application focuses on the role of democratic political institutions in reducing the likelihood of civil war onset. Democracy is a fundamental concept when modeling the quality of governance, but drawing inferences about it's effect represents a straightforward example of the multiple causes setting. In particular, democracy cannot be measured as a single unambiguous feature – instead it is a confluence of many conceptually related by empirically distinct features describing different aspects of a system of governance. The causal effect of democracy on outcomes like conflict initiation is typically measured using a dimension reduction of the features representing the individual institutions [Treier and Jackman, 2008]; however, such strategies have led to conflicting results about the importance of democracy for political stability [Vreeland, 2008, Fearon and Laitin, 2003]. Consequently, the role of democratic political institutions in reducing civil war onset represents a useful case for comparing latent trait models with with the MCSE.

Specifically, we used a linear model with fixed effects for the country and year for both $\hat{P}(\mathcal{T}' = \mathcal{T}_q^{\text{Max}} \cap \mathcal{T}'' = \mathcal{T}_q^{\text{Min}})$ and $\hat{\tau}(\mathcal{T}', \mathcal{T}'')$ and measured democratic political institutions using 111 features describing the system of governance present in a country in the Varieties of Democracy Dataset (V-Dem).[9] The dots and confidence intervals on the left in Figure 3 show the estimates for $\text{MCSE}_q$ quantifying the effect of these political institutions on civil war onset for many different values of $q$. In particular, they suggest that countries with one of the 10% most conflict reducing sets of institutions have roughly a 1.2% lower risk of civil war than countries with some of the 10% most conflict inducing institutions. The dots on the right instead represent estimates for the $\text{MCSE}_q$ using just the V-Dem polyarchy indicator, which is a standard measure of Democracy in the political science literature. The estimates for MCSE are larger than those achieved using the univariate model,

---

[9]See appendix B.1 for more details on these models.

Figure 3: The Causal Effect of Democratic Political Institutions on the Probability of Civil War Onset

**Note:** The red dots identify estimates for the $\text{MCSE}_q$ made using the methodology outlined in this paper and represent the combined influence of many different democratic institutions together. The blue dots instead represent the influence of just a univariate latent trait produced by the maintainers of the V-Dem Dataset that is frequently used to model democracy.
**Note 2:** Confidence intervals adjusted for clustering by country.

suggesting that the the MCSE can successfully recover causal effects that standard latent variable approaches cannot.

## Conclusion

Non-parametric estimation techniques and high dimensional datasets increasingly confront researchers with estimates for a huge number of distinct causal estimands. While the capacity to fit such models represents tremendous progress for the estimation and computational techniques that support them, scientific theories rarely make predictions about such a large number of distinct parameters. In this article, we propose a framework for making sense of such model outputs by focusing on the maximum causal contrast between two sets of a researcher specified size $q$. We also develop an estimator for this estimand that is consistent, conservative in finite samples, and asymptotically normal. While the estimator is developed with the many causes and treatment effect heterogeneity settings in mind, the framework is extremely flexible and could be extended to a myriad of other causal qauntities of interest, speaking to its wide applicability and utility for applied researchers. While a single causal estimand will never replace the kind of careful synthetic and analysis of individual causal variables should accompany the study of any complex phenomenon, we believe the MCSE will be a useful tool in a wide array of scientific disciplines.

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
