# A  Restoring Efficiency With Cross Fitting

The estimator described in Section 4 is inefficient because it only uses a subset of the data for each of the two steps. Efficiency can be improved by rotating the roles of the splitting and estimation sets, repeating the estimation procedure and then averaging the results – a procedure that Chernozhukov et al. [2017] term cross-fitting. Formally, we now assume that the researcher has randomly split the data into $K$ folds and let $\widehat{\text{MCSE}}_q^{(k)}$ denote the estimate for $\text{MCSE}_q$ generated using fold $k$ as $\mathcal{S}^{\text{Prob}}$ and the remaining folds as $\mathcal{S}^{\text{Est}}$. The crossfit estimator then takes the form:

$$\widehat{\text{MCSE}}_q^{\text{CF}} = \frac{1}{K} \sum_{k=1}^{K} \widehat{\text{MCSE}}_q^{(k)}$$

Because this estimator is an average of conservative and consistent estimators for $\text{MCSE}_q$, it will itself be a conservative and consistent estimator for the $\text{MCSE}_q$ as long as the assumptions introduced in Section 4.2 are maintained. By the same logic, it will also be asymptotically normal if the assumptions needed for Proposition 4 are also maintained.

Because this estimator reuses the same data, estimates $\widehat{\text{MCSE}}_q^{(k)}$ for different choices of $k$ will not be independent.[10] Consequently, I use the following conservative variance estimator:

**Proposition 5.** *Let,*

$$\widehat{Var}\left(\widehat{\text{MCSE}}_q^{CF}\right) = \frac{1}{K} \sum_{k=1}^{K} \widehat{Var}\left(\widehat{\text{MCSE}}_q^{(k)}\right)$$

*where $\widehat{Var}\left(\widehat{\text{MCSE}}_q^{(k)}\right)$ is the bootstrap variance estimator for $\widehat{\text{MCSE}}_q^{(k)}$ discussed in Section 4.3.*

*Then,*

$$\mathbb{E}\left(\widehat{Var}\left(\widehat{\text{MCSE}}_q^{CF}\right)\right) \leq Var\left(\widehat{\text{MCSE}}_q^{CF}\right)$$

*Proof.* Conservatism of this estimator follows trivially from the validity of the boostrap variance estimator demonstrated in Section 4.3 and one application of the Cauchy-Schwarz inequality. $\square$

The bias of this variance estimator will be minimized when there are only 2 folds. Since performance of $\widehat{\text{MCSE}}_q$ is not otherwise impacted by the number of folds, most applications will be best served by setting $K$ equal to 2.

# B  Additional Experimental Details

## B.1  $\widehat{\text{MCSE}}_q$ Estimator Details

Both experiments involve using a linear model for both $\hat{P}(\mathcal{T}' = \mathcal{T}_q^{\text{Max}} \cap \mathcal{T}'' = \mathcal{T}_q^{\text{Min}})$ and $\hat{\tau}(\mathcal{T}', \mathcal{T}'')$. Specifically, the estimator for $\hat{P}(\mathcal{T}' = \mathcal{T}_q^{\text{Max}} \cap \mathcal{T}'' = \mathcal{T}_q^{\text{Min}})$ first fits the following regression using only observations $i \in \mathcal{S}^{\text{Prob}}$:

$$\mathbb{E}\left(\widehat{Y_i|\mathbf{t}_i}, x_i\right) = \mathbf{t}_i'\hat{\beta} + x_i'\hat{\gamma}$$

Where $\beta$ and $\gamma$ are regression coefficients and $\mathbf{t}_i$, $x_i$, and $Y_i$ are defined in the same way as in Section 2.3. We then construct estimates for $\hat{P}(\mathcal{T}' = \mathcal{T}_q^{\text{Max}} \cap \mathcal{T}'' = \mathcal{T}_q^{\text{Min}})$ by first monte carlo sampling $B$ values of $\beta^{(b)}$ and $\gamma^{(b)}$ for $b = 1 \ldots B$ such that:

---

[10]Indeed, simulations demonstrate that the correlation between estimates with different values of $K$ can be as high as .9.

$$\begin{bmatrix} \beta^{(b)} \\ \gamma^{(b)} \end{bmatrix} \sim \mathcal{N}\left( \begin{bmatrix} \hat{\beta} \\ \hat{\gamma} \end{bmatrix}, \hat{\sigma}^2 Q^{-1} \right)$$

where $Q = [T' \quad X'] \begin{bmatrix} T \\ X \end{bmatrix}$ where the matrices $T$ and $X$ are defined such that each row $i$ in $T$ contains vector $\mathbf{t}_i$ and each row $i$ in $X$ contains vector $x_i$. Predictions $\mathcal{T}_q^{\text{Max}(b)}$ and $\mathcal{T}_q^{\text{Min}(b)}$ are then constructed as:

$$\mathcal{T}_q^{\text{Max}(b)} = \arg\max_{\mathcal{T}' \in \mathcal{T}_q} \sum_{i \in \mathcal{S}^{\text{Prob}}} \frac{\mathbb{1}\{\mathbf{t}_i \in \mathcal{T}'\} \mathbf{t}_i \beta^{(b)}}{\mathbb{1}\{\mathbf{t}_i \in \mathcal{T}'\}}$$

and

$$\mathcal{T}_q^{\text{Min}(b)} = \arg\min_{\mathcal{T}' \in \mathcal{T}_q} \sum_{i \in \mathcal{S}^{\text{Prob}}} \frac{\mathbb{1}\{\mathbf{t}_i \in \mathcal{T}'\} \mathbf{t}_i \beta^{(b)}}{\mathbb{1}\{\mathbf{t}_i \in \mathcal{T}'\}}$$

Finally, $\hat{P}(\mathcal{T}' = \mathcal{T}_q^{\text{Max}} \cap \mathcal{T}'' = \mathcal{T}_q^{\text{Min}})$ is constructed from these monte carlo draws as:

$$\hat{P}(\mathcal{T}' = \mathcal{T}_q^{\text{Max}} \cap \mathcal{T}'' = \mathcal{T}_q^{\text{Min}}) = \sum_{b=1}^{B} \frac{\mathbb{1}\{\mathcal{T}_q^{\text{Max}(b)} = \mathcal{T}'\} \mathbb{1}\{\mathcal{T}_q^{\text{Min}(b)} = \mathcal{T}''\}}{B}$$

Estimates for $\hat{\tau}(\mathcal{T}', \mathcal{T}'')$ on the other hand are constructed by first subsetting to just the elements of $\mathcal{S}^{\text{Est}}$ that are contained in either $\mathcal{T}'$ or $\mathcal{T}''$ and then using those observations to fit the following linear regression:

$$\mathbb{E}\,\widehat{(Y_i|\mathbf{t}_i, x_i)} = \mathbb{1}\{\mathbf{t}_i \in \mathcal{T}'\}\hat{\tau}(\mathcal{T}', \mathcal{T}'') + x_i'\hat{\gamma}$$

where $\hat{\tau}(\mathcal{T}', \mathcal{T}'')$ and $\hat{\gamma}$ are estimated via OLS.

In the synthetic data experiment, $\mathbf{t}_i$ is unconfounded by construction, so $x_i$ is a null set. In the democratic institutions application on the other hand, $x_i$ includes a dummy variable indicating whether country $i$ was in a state of civil war in any of the previous four years and a set of dummy variables corresponding to the country and year fixed effects. Note that such fixed effects estimators are a common approach to causal inference when using panel data in this type and can recover estimates of causal effects under assumptions that, while certainly debatable, are relatively common in the social sciences [Wooldridge, 2010]. In both cases, we use 100 monte carlo draws to estimate $\hat{P}(\mathcal{T}' = \mathcal{T}_q^{\text{Max}} \cap \mathcal{T}'' = \mathcal{T}_q^{\text{Min}})$ and 100 bootstrap iterations to estimate the confidence intervals in the democratic institutions experiment.

## B.2  Details on Implementation of the PCA Estimator

For the PCA estimator, we began by conducting a principal components analysis of the set $\{\mathbf{t}_i : i \in \mathcal{S}\}$ to extract the corresponding value of the first principal component $z_i$ for all units $i$. We then estimated the conditional expectation of $Y_i$ given $z_i$ using the following linear regression:

$$\mathbb{E}\,\widehat{(Y_i|z_i)} = z_i \beta$$

where $\beta$ is the regression coefficient estimated using OLS. We then estimate $\text{MCSE}_q$ as:

$$\sum_{i \in \mathcal{S}} \frac{\mathbb{E}\,\widehat{(Y_i|z_i)}\mathbb{1}\{\mathbb{E}\,\widehat{(Y_i|z_i)} > \mathcal{Q}_{1-q}\}}{\mathbb{1}\{\mathbb{E}\,\widehat{(Y_i|z_i)} > \mathcal{Q}_{1-q}\}} - \sum_{i \in \mathcal{S}} \frac{\mathbb{E}\,\widehat{(Y_i|z_i)}\mathbb{1}\{\mathbb{E}\,\widehat{(Y_i|z_i)} < \mathcal{Q}_q\}}{\mathbb{1}\{\mathbb{E}\,\widehat{(Y_i|z_i)} < \mathcal{Q}_q\}}$$

Where $\mathcal{Q}_{1-q}$ and $\mathcal{Q}_q$ represent the $1-q$ and $q$ quantiles of $\{\mathbb{E}\,\widehat{(Y_i|z_i)} : i \in \mathcal{S}\}$

Figure 4: Simulation Root Mean Squared Error

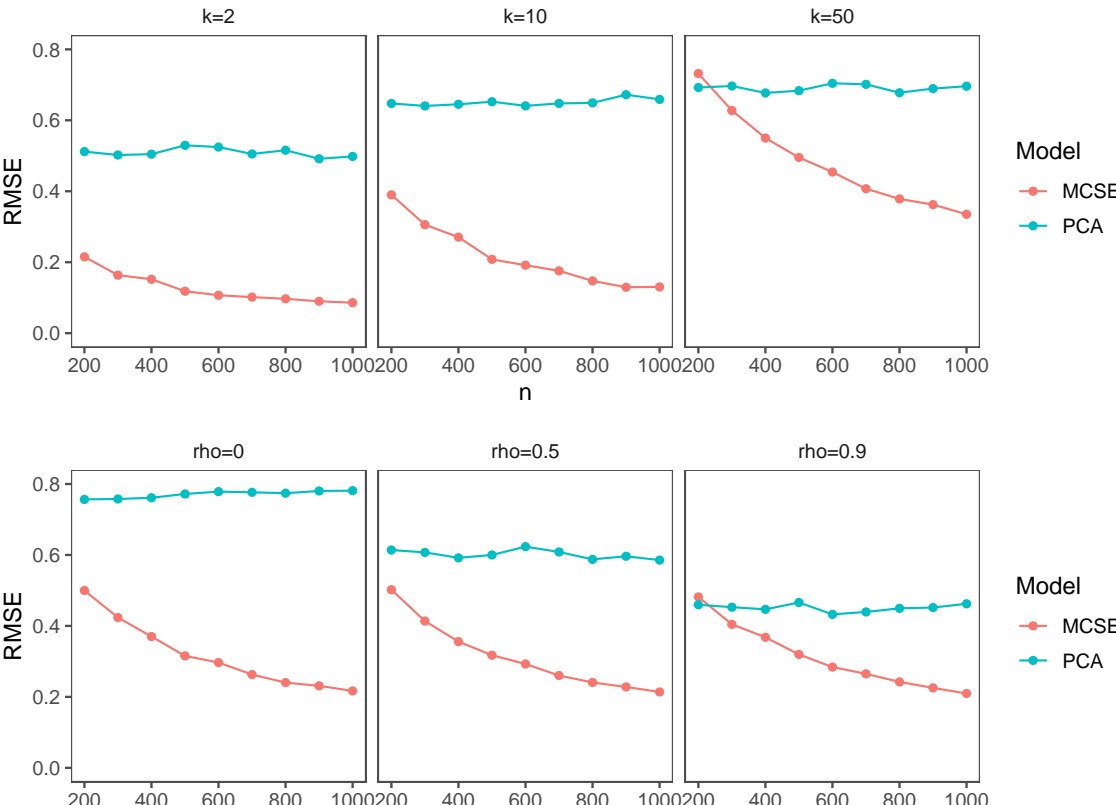

## B.3 Root Mean Squared Error in Synthetic Data

Figure 4 visualizes the root mean squared error (RMSE) for the same two estimators using the same simulation set up. Overall the RMSE shows the same general pattern as the bias, unless $n$ is very small, the MCSE estimator outperforms the PCA estimator, suggesting that there are considerable gains in efficiency from modeling the full effect of the many causes, even when the many causes are highly correlated.

Figure 5: False Positive Rate in Synthetic Data

## B.4 False Positive Rate in Synthetic Data

We also validated the performance of our variance estimator on simulated data. Because the standard errors are conservative (see the discussion of the cross-fit estimator in Section A), we focused on the control over the false positive rate when the true MCSE is equal to zero. Specifically, null set up used the same simulation structure as those in the main paper, except that $\mu_i$ is always zero. Figure 5 visualizes the results of this analysis, when the threshold for statistical significance is assumed to be .05. Here each dot represents the false positive rate in 300 simulations (averaging over the three different value of $\rho$). Overall, the results are encouraging. The false positive rate is consistently low, and usually much below .05, as we would expect from a conservative variance estimator.

## C Proofs

### C.1 Proof for Proposition 1

*Proof.* Note, by construction, for any $\mathcal{T}', \mathcal{T}'' \in \mathcal{T}_q$ :

$$\text{MCSE}_q = \max_{\mathcal{T}',\mathcal{T}''\in\mathcal{T}_q} \tau(\mathcal{T}', \mathcal{T}'') = \tau\left(\mathcal{T}_q^{\text{Max}}, \mathcal{T}_q^{\text{Min}}\right) \geq \tau(\mathcal{T}', \mathcal{T}'')$$

Therefore, from Assumption 1 and the assumption that $\mathbb{E}\left(\hat{\tau}(\mathcal{T}', \mathcal{T}'')\right) \leq \tau(\mathcal{T}', \mathcal{T}'')$ introduced in the proposition statement

$$\begin{aligned}
\text{MCSE}_q &\geq \sum_{\mathcal{T}',\mathcal{T}''\in\mathcal{T}_q} \mathbb{E}\left(\hat{P}(\mathcal{T}' = \mathcal{T}_q^{\text{Max}} \cap \mathcal{T}'' = \mathcal{T}_q^{\text{Min}})\right) \tau(\mathcal{T}', \mathcal{T}'') \\
&= \sum_{\mathcal{T}',\mathcal{T}''\in\mathcal{T}_q} \mathbb{E}\left(\hat{P}(\mathcal{T}' = \mathcal{T}_q^{\text{Max}} \cap \mathcal{T}'' = \mathcal{T}_q^{\text{Min}})\right) \mathbb{E}\left(\hat{\tau}(\mathcal{T}', \mathcal{T}'')\right) \\
&= \mathbb{E}\left(\widehat{\text{MCSE}_q}\right)
\end{aligned}$$

$\square$

### C.2 Proof of Theorem 2

*Proof.* In the following proof, I use super-scripting by $(n)$ to denote a quantity that is dependent on the sample size and has been estimated using a sample of size $n$. Turning to the proof, first consider the case when $\forall \mathcal{T}', \mathcal{T}'' \subseteq \mathcal{T}$:,

$$\hat{P}^{(n)}(\mathcal{T}' = \mathcal{T}_q^{\text{Max}} \cap \mathcal{T}'' = \mathcal{T}_q^{\text{Min}}|\mathcal{S}^{\text{Prob}}) \xrightarrow[n\to\infty]{p} \mathbb{1}\{\mathcal{T}' = \mathcal{T}_q^{\text{Max}}\}\mathbb{1}\{\mathcal{T}' = \mathcal{T}_q^{\text{Max}}\}$$

and,

$$\hat{\tau}^{(n)}(\mathcal{T}', \mathcal{T}'') \xrightarrow[n \to \infty]{p} \tau(\mathcal{T}', \mathcal{T}'')$$

Then,

$$
\begin{aligned}
\widehat{\text{MCSE}}_q^{(n)} &= \sum_{\mathcal{T}, \mathcal{T}'' \in \mathcal{T}_q} \hat{P}^{(n)}(\mathcal{T}' = \mathcal{T}_q^{\text{Max}} \cap \mathcal{T}'' = \mathcal{T}_q^{\text{Min}} | \mathcal{S}^{\text{Prob}}) \hat{\tau}^{(n)}(\mathcal{T}', \mathcal{T}'') \\
&\xrightarrow[n \to \infty]{p} \sum_{\mathcal{T}, \mathcal{T}'' \in \mathcal{T}_q} \mathbb{1}\{\mathcal{T}' = \mathcal{T}_q^{\text{Max}}\} \mathbb{1}\{\mathcal{T}' = \mathcal{T}_q^{\text{Max}}\} \hat{\tau}^{(n)}(\mathcal{T}', \mathcal{T}'') \\
&\xrightarrow[n \to \infty]{p} \tau(\mathcal{T}_q^{\text{Max}}, \mathcal{T}_q^{\text{Min}}) \\
&= \text{MCSE}_q
\end{aligned}
$$

The proof for almost sure convergence is identical to that for convergence in probability, however, convergence in probability is replaced with almost sure convergence in all cases.

$\square$

## C.3 Proof for Proposition 3

Arbitrarily index every ordered pair $(\mathcal{T}', \mathcal{T}'') \in \mathcal{T}_q$ with $1 \dots J$ and let $w_i$, $d_i$, and $\hat{\tau}_i$ denote the corresponding random variables $w(\mathcal{T}', \mathcal{T}'')$, $d(\mathcal{T}', \mathcal{T}'')$, and $\hat{\tau}(\mathcal{T}', \mathcal{T}'')$. Further, let $\epsilon_i = \hat{\tau}_i - \tau$. First note that by the tower property,

$$
\begin{aligned}
\mathbb{E}\left(\left(\sum_{i=1}^{J} d_i \epsilon_i\right)^2\right) &= \mathbb{E}\left(\mathbb{E}\left(\left(\sum_{i=1}^{J} d_i \epsilon_i\right)^2 \Big| d\right)\right) \\
&= \mathbb{E}\left(\mathbb{E}\left(\left(\sum_{i=1}^{J} \mathbb{1}\{d_i = 1\}\epsilon_i\right)^2 \Big| d\right)\right) \\
&= \mathbb{E}\left(\left(\sum_{i=1}^{J} \mathbb{E}\left(\mathbb{1}\{d_i = 1\}\epsilon_i^2\right) \Big| d\right)\right) \\
&= \sum_{i=1}^{J} \mathbb{E}\left(\mathbb{1}\{d_i = 1\}\right) \mathbb{E}\left(\epsilon_i^2\right) \\
&= \sum_{i=1}^{J} \mathbb{E}\left(w_i\right) \mathbb{E}\left(\epsilon_i^2\right)
\end{aligned}
$$

And by the Cauchy Schwarz inequality,

$$\mathbb{E}\left(\left(\sum_{i=1}^{J} w_i \epsilon_i\right)^2\right) = \mathbb{E}\left(\sum_{i,j=1}^{J} w_i \epsilon_i w_j \epsilon_j\right)$$

$$\leq \mathbb{E}\left(\sum_{i=1}^{J} w_i^2 \epsilon_i^2\right)$$

$$\leq \sum_{i=1}^{J} \mathbb{E}\left(w_i^2\right) \mathbb{E}\left(\epsilon_i^2\right)$$

$$\leq \sum_{i=1}^{J} \mathbb{E}\left(w_i\right) \mathbb{E}\left(\epsilon_i^2\right)$$

where the final inequality holds because $w_i$ is between 0 and 1.

### C.4    Proof for Proposition 4

*Proof.* As is the case in Appendix C.2, I use super-scripting by $(n)$ to denote a quantity that is dependent on the sample size and has been estimated using a sample of size $n$.

For any $\mathcal{T}', \mathcal{T}'' \in \mathcal{T}_q$ let $v_i^{(n)} = f_i^{(n)}(Z, \mathcal{T}', \mathcal{T}'')Y_i - \mathbb{E}\left(f_i^{(n)}(Z, \mathcal{T}', \mathcal{T}'')Y_i\right)$ where $f_i^{(n)}()$ and $Z$ have the same definition as in assumption 2. Also, to simplify notation, let $p^{(n)} = \hat{P}(\mathcal{T}' = \mathcal{T}_q^{\text{Max}} \cap \mathcal{T}'' = \mathcal{T}_q^{\text{Min}})$.

Neumann [2013] shows that a sufficient condition for $\sum_{i \in \mathcal{S}^{\text{Est}}} p^{(n)} v_i^{(n)}$ to be asymptotically normal is that along with satisfying the lindberg-feller condition and having finite second moments (which are both assumed in the statement of the proposition), for any $i$, $j$, and any measurable function $l$ such that $||l||_\infty \leq 1$,

$$\text{Cov}(l(p^{(n)} v_{-j}^{(n)}) p^{(n)} v_i^{(n)}, p^{(n)} v_j^{(n)} | Z) = 0$$

and

$$\text{Cov}(l(p^{(n)} v_{-i,j}^{(n)}), p^{(n)} v_i^{(n)} p^{(n)} v_j^{(n)}) = 0$$

where $p^{(n)} v_{-j}^{(n)}$ denotes the vector of outcomes except for $v_i^{(n)}$ and $p^{(n)} v_{-i,j}^{(n)}$ denotes the same, but exempting both units $i$ and $j$. To simplify the notation in the following derivations, I omit the explicit conditioning on $Z$, but all expectations and covariance should be understood as being conditional on $Z$. However, note the assumption that $Y_i \perp\!\!\!\perp Y_j$ for all $i \neq j$ implies that conditional on $Z$, $v_i^{(n)} \perp\!\!\!\perp v_j^{(n)}$

Now, considering the first condition,

$$\text{Cov}(l(p^{(n)} v_{-j}^{(n)}) p^{(n)} v_i^{(n)}, p^{(n)} v_j^{(n)}) = \mathbb{E}\left(l(p^{(n)} v_{-j}^{(n)}) p^{(n)} v_i^{(n)} p^{(n)} v_j^{(n)}\right) - \mathbb{E}\left(l(p^{(n)} v_{-j}^{(n)}) p^{(n)} v_i^{(n)}\right) \mathbb{E}\left(p^{(n)} v_j^{(n)}\right)$$

$$= \mathbb{E}\left(l(p^{(n)} v_{-j}^{(n)}) p^{(n)} v_i^{(n)} p^{(n)}\right) \mathbb{E}\left(v_j^{(n)}\right) - \mathbb{E}\left(l(p^{(n)} v_{-j}^{(n)}) p^{(n)} v_i^{(n)}\right) \mathbb{E}\left(p^{(n)}\right) \mathbb{E}\left(v_j^{(n)}\right)$$

$$= \mathbb{E}\left(v_j^{(n)}\right) \left(\mathbb{E}\left(l(p^{(n)} v_{-j}^{(n)}) p^{(n)} v_i^{(n)} p^{(n)}\right) - \mathbb{E}\left(l(p^{(n)} v_{-j}^{(n)}) p^{(n)} v_i^{(n)}\right) \mathbb{E}\left(p^{(n)}\right)\right)$$

$$= 0 \left(\mathbb{E}\left(l(p^{(n)} v_{-j}^{(n)}) p^{(n)} v_i^{(n)} p^{(n)}\right) - \mathbb{E}\left(l(p^{(n)} v_{-j}^{(n)}) p^{(n)} v_i^{(n)}\right) \mathbb{E}\left(p^{(n)}\right)\right)$$

$$= 0$$

Next, the second condition,

$$\mathrm{Cov}(l(p^{(n)}v_{-i,j}), p^{(n)}v_i p^{(n)}v_j) = \mathbb{E}\left(l(p^{(n)}v_{-i,j})p^{(n)}v_i p^{(n)}v_j\right) - \mathbb{E}\left(l(p^{(n)}v_{-i,j})\right)\mathbb{E}\left(p^{(n)}v_i p^{(n)}v_j\right)$$

$$= \mathbb{E}\left(l(p^{(n)}v_{-i,j})\left(p^2\right)^{(n)}\right)\mathbb{E}(v_i)\mathbb{E}(v_j) - \mathbb{E}\left((p^{(n)}v_{-i,j})\right)\mathbb{E}\left(\left(p^{(n)}\right)\right)^2\mathbb{E}\left(v_i^{(n)}\right)\mathbb{E}\left(v_j^{(n)}\right)$$

$$= \mathbb{E}\left(v_i^{(n)}\right)\mathbb{E}\left(v_j^{(n)}\right)\left(\mathbb{E}\left((p^{(n)}v_{-i,j})\left(p^{(n)}\right)^2\right) - \mathbb{E}\left((p^{(n)}v_{-i,j}^{(n)})\right)\mathbb{E}\left(\left(p^{(n)}\right)^2\right)\right)$$

$$= 0\left(\mathbb{E}\left((p^{(n)}v_{-i,j}^{(n)})\left(p^{(n)}\right)^2\right) - \mathbb{E}\left((p^{(n)}v_{-i,j}^{(n)})\right)\mathbb{E}\left(\left(p^{(n)}\right)^2\right)\right)$$

$$= 0$$

So, we conclude that $\sum_{i \in \mathcal{S}^{\mathrm{Est}}} p^{(n)}v_i^{(n)} \xrightarrow[n \to \infty]{D} \mathcal{N}\left(0, \mathrm{Var}\left(\sum_{i \in \mathcal{S}^{\mathrm{Est}}} v_i^{(n)}\right)\right)$. Since the sum of normal random variables is also normally distributed, we then conclude that, $\widehat{\mathrm{MCSE}}_q^{(n)} = \sum_{\mathcal{T}, \mathcal{T}'' \in \mathcal{T}_q} \hat{P}^{(n)}(\mathcal{T}' = \mathcal{T}_q^{\mathrm{Max}} \cap \mathcal{T}'' = \mathcal{T}_q^{\mathrm{Min}} | \mathcal{S}^{\mathrm{Prob}})\hat{\tau}^{(n)}(\mathcal{T}', \mathcal{T}'')$ will also be asymptotically normal, completing the proof. $\square$