# OpenReview forum: "Answering Complex Causal Queries With the Maximum Causal Set Effect"
_NeurIPS.cc/2021/Conference — NeurIPS 2021 Poster_

### Official Review · Reviewer_XkeP · 2021-07-16

**Rating:** 8
**Confidence:** 4

**Summary:**

### 1. Summary

The paper introduces a novel estimator for the multiple causes / moderators/ mediatiors scenario within causal inference processes.

The paper claims a threefold contribution, namely

(i) to introduce the notion of "complex causal queries"

(ii) to introduce a novel estimand (MCSE)

(iii) to introduce a related estimator

**Limitations And Societal Impact:**

Limitations and societal impact are adressed.

**Main Review:**

### 2. Rationale for the score

Overall the paper follows a good structure and shows a adequate style of writing. A good structure with highlighted Lemmata and definitions is used. The choice of the problem is interesting and up-to-date. The lack of previous work on the topic convinces me that the paper should be accepted.

### 3. Positive aspects

- the problem is nicely introduced

- the accompanying expirements with synthetic and real data are fitting to the problem description

- using the algorithm on socio-political data (here civil war onset), is a very interesting application and fits the outlook regarding broader impacts

-the overall structure is well formulated and the seperate handling of point estimation and interval estimation properties seems advisable

### 4. Negative aspects

- while examples are provided within the theoretical parts of the paper, I feel these could be better integrated to serve as illustration of the concepts explained.

-I would suggest to highlight the claimed contributions of the paper more clearly and also refer to these in the conclusion.



**Time Spent Reviewing:**

4

---

> ### Author Response · Authors · 2021-08-05
> **Response to Reviewer XkeP**
>
> We thank the reviewer for their positive comments and helpful suggestions. We will work to strengthen the conclusion and better integrate our examples into the paper, as the reviewer advises.

---

### Official Review · Reviewer_jFfp · 2021-07-17

**Rating:** 5
**Confidence:** 2

**Summary:**

This paper describes and procedure for estimating causal effects in the setting where there are many treatments of interest and the analyst wishes to summarize the relationship between those many treatments and the outcome. The authors define an estimand, the maximum causal set effect, which describes the treatment effect implied by intervening on the sets of treatment variables with different values. The authors provide theoretical results demonstrating the convergence/asymptotic normality of their proposed estimator for this effect using sample splitting. They then apply their estimator to synthetic and real-world data.

**Ethical Concerns:**

I don't think my above point rises to the status of "needs ethical review". It's unlikely this paper will immediately see wide adoption. The authors, rather, should carefully ponder what exactly their broader contribution is as they revise.

**Limitations And Societal Impact:**

See above comments about the efficacy of summarizing complex research. I don't think the authors make a strong case. In fact, they explicitly allude to negative examples (in their over-the-limit Broader Impacts section) like IQ. It's well-understood that IQ is an inappropriate tool that has been used in the perpetuation of racist and sexist practices (see, e.g. Weiten 2016). Intelligence is a multi-faceted characteristic of a person that CANNOT be boiled down to one number and therefore SHOULD not. I am concerned the authors are falling into the standard NeurIPS trap of saying "this seems like an interesting technical idea" and casting it as ground breaking for applied science. It has the potential for damage and the potential for benefit is, as I've stated, limited and unclear.

**Main Review:**

I found the proposed methodology in this paper to be modestly interesting and, once you get past some notational opaqueness, the backing theory appears to be sound (i.e. I can tell what the authors are getting at). There are some rather fundamental issues with the exposition and motivation that make it unclear why exactly this approach is "necessary". Perhaps some of these issues can be cleared up through the review process but for now I think the paper warrants major revisions.

I'll also add that a full paragraph (Broader Impacts section, about 15 lines) spills over onto a 10th page.

Detailed comments follow:

**Motivation** -- Throughout the paper I struggled to understand what is fundamentally wrong with what the authors cast as the woeful status quo. The authors point to applied researchers as being unable to properly characterize the effects of many variables on one outcome. I think this is...sorta the point of applied science? The world is complex and often _shouldn't_ be characterized in terms of univariate summaries (take IQ as an exceedingly good example of something that does an exceedingly bad job of capturing something multifaceted, intelligence). This has parallels with uncertainty quantification or set identifiability. People might find it upsetting or dissatisfying that there isn't _one_ right answer. The reality, though, is that the complexity in the modeling/statistics exists because the problems themselves are complex.

Following that point, the authors don't really make clear why it's infeasible (or not desirable) to intervene on all the treatments of interest with separate treatment settings for the sake of comparison, or, alternatively, intervene on each treatment successively to get a characterization of each treatment's causal relationship with the outcome.

"Techniques developed in the context of simple causal queries cannot be used to answer complex ones" -- why not? Is there a citation? What is the "challenge" to which the authors refer? Statistical? Interpretation?

**Multiple cause vs. multiple moderators** -- It's not clear how the two graphs in Fig. 1 are different. In a non-parametric causal model, it's standard for there to be interaction effects so I don't see the point of the distinction.

**Notation/Assumptions** -- What is meant by "treatment type" (line 103)? Is it the value of the treatments? Are they all binary, continuous, other?

Up to section 3, caligraphy{T} isn't clearly defined. It appears to be a set of treatment variables which are (implicitly?) intervened upon. Is that understanding correct? This should be stated explicitly before it's used in downstream definitions

The definitions of Tmax and Tmin aren't clear. It appears the authors mean "Tmax and Tmin are the settings of the treatment variables T such that the difference in average outcome Y under Tmax and under Tmin is maximal". Even the definition on line 135 doesn't quite make this explicit.

The interpretations of Tmax and Tmin are further confused by the authors' example of picking the "10% best-performing casts" vs. "10% worst-performing casts". It seems the idea is to "peak" at Y to determine the relative effects of various treatment variables. To determine the top 10%, I don't see how the analyst could avoid an exhaustive "intervene on everything successively" approach like the one I alluded to above.

**"A basic result in the Q-learning literature** -- citation? Is it really basic?

**ML techniques for estimating the probability chunk** -- The authors claim ML techniques are incapable of providing probabilistic estimates for Phat(Tmax = T' \cap Tmin = T''). I'm not sure I agree with this. There are several out-of-the-box multi-class classification models that can provide probability estimates. For instance, RFs inherently are multi-class and sklearn's implementation provides out-of-box probability predictions.

**Experiments** -- what exactly is the "intervention" here? Are you picking a specific q and then finding sets to maximize Y according to that q?

"In particular, they suggest that countries with one of the 10% most conflict reducing institutions have..." -- similar to above comment, this seems a bit like the tail wagging the dog. The authors are explicitly looking at the outcome to find effects of interest and then casting it as surprising that high "performing" interventions do "better" than low "performing" interventions.

**Time Spent Reviewing:**

6

---

> ### Author Response · Authors · 2021-08-05
> **Response to Reviewer jFfP**
>
> We thank the reviewer for their thought provoking and useful comments. The reviwer's primary concerns center on the motivation and possible negative social impacts of our results, and our response is primarily directed at these concerns. The reviewer also raises some questions about how our estimator functions when discussing the experiments, which we consider at the end of our response.
>
> **Motivation**
> The reviewer's primary concern focuses on the motivation for the proposed estimand. Specifically, they argue that the world is complex and that researchers should not seek a "univariate summary" of the effect of a bundled treatment. Instead, they propose 1) intervening on "all treatments of interest" or 2) "intervening on each treatment sequentially". The principal challenge with such approaches in the face of complex causal queries is that they require interpreting causal effects associated with a huge number of distinct interventions. Considering the democracy application, the first strategy would require considering the change in the probability of civil war onset between countries with every possible combination of democratic political institutions while the second would require considering the change in the probability of civil war onset associated with changing each democratic political institution individually while holding all else constant. Unfortunately, the dataset used in the democracy application includes 128 variables and either of the two strategies would require qualitatively interpreting a massive number of causal estimates to reach a conclusion about the causal effect of democracy on the likelihood of civil war onset. This highlights what we view as the principal problem posed by complex causal queries: when causal queries are stated in terms of many variables, the massive number of causal estimates that conventional approaches provide will be uninterpretable.
>
> While the reviewer recognizes the challenge of interpreting such a large number of causal estimates, they worry that researchers might use the MCSE to avoid the messy work of synthesis and theory building that an analysis of any complex phenomenon requires. In broad strokes, we are sympathetic to this criticism. Indeed, we see the MCSE as a complement to, rather than a replacement for, more conventional statistical approaches and will clarify this intention when revising the manuscript. For example, an applied researcher studying the effect of democracy on the probability of civil war onset would certainly want to discuss the effect of a few democratic political institutions in particular (for example, whether a country has fair and free elections, whether freedom of speech is respected, etc.) along with using the MCSE to quantify the effect of democratic political institutions on the probability of civil war onset in general.
>
> Nonetheless, we believe there is no shortage of scientific questions that could benefit from the use of the MCSE. For example, identifying the degree of heritability of a particular disease from genomics data will require modeling the effect of an organism's entire genome on some phenotype, and the focus of current techniques for genome-wide association studies on individual alleles is recognized as a major shortcoming when attempting to explain complex diseases (Tam et al 2019). Similarly, researchers have relied on dimension reduction techniques to model causal phenomena as disparate as the effect of democracy on civil war onset (Trier and Jackman 08), social variables on the likelihood of premature death (Kolak et al20), and the effect of radiation therapy on cancer survival (Nabi et al 20). Such dimension reduction techniques represent an alternative approach to producing a univariate summary of the causal effect of many treatments on some outcome, and their prevalance suggests that applied researchers are accustomed to stating causal queries in terms of bundles of conceptually related variables.
>
> **Social Impact**
> The reviewer also raises important concerns about the potential societal impact of introducing the MCSE; however, we believe that such harms really stem from the variables that form the contents of the bundle rather than the choice to analyze those variables as a bundle itself. For example, would it really be less racist to argue that intelligence predicts income by considering the questions that compose IQ scores separately instead of as a bundle? We feel strongly that the harms of IQ testing stem from equivalating questions that principally measure white cultural knowledge with general intelligence rather than from the notion of bundled treatments more generally. Such piecemeal approaches might even amplify these harms relative to using the MCSE -- a researcher attempting to make sense of a huge number of causal estimates could easily be tempted to focus on a few extreme results that confirm their own biases and predispositions.
>
> We also believe that our methodology will have much utility for answering socially relevant questions as well. Race in particular has been identified as an example of a bundled treatment (Sen and Wasow 16). Indeed, constructivist theories of race emphasize that it does not exist as a binary trait and is really composed of bundles of related traits (e.g. skin tone, dialect, socio-economic status, etc.) that may or may not co-occur in a particular individual. Focusing on race as a binary trait, as is currently done in most statistical analyses, may lead to researchers understating the importance of race as a causal variable on many important outcomes.
>
> **Experiments**
> The reviwer asks about the interpretation of the experiments, specifically asking whether we are "picking a specific q and then finding sets to maximize Y according to that q?" This interpretation is correct -- our methodology assumes that the researcher has specified the value of q. This is mentioned on line 132, but we will revise the experiments section to reiterate this detail.
>
> In light of this definition, the reviwer wonders whether it is really "surprising that high 'performing' interventions do 'better' than low 'performing' interventions?" In essense, this concern provides the intuition for why a split-sample estimation approach is needed to avoid a positively biased estimator. Indeed, a single sample estimator for the MCSE would be badly biased and provide a positive estimate for the MCSE regardless of the estimand's true value. This behavior regarding the maximum of a set of random variables is well known in statistics and is alternately referred to as "selection bias", "the winner's curse", or "regression to the mean." It also provides motivation for double q-learning approaches which are discussed in Hasselt 2010.
>
> Our split-sample approach is able to overcome this bias. By using a different (and therefore independent) subset of the data to identify the minimum and maximum causal sets than is used to estimate their effects, we are able to provide an estimator with a downward bias. Our main technical results are aimed at showing the conservatism of this split-sample estimator and deriving a valid framework for testing the null hypothesis that the true MCSE is zero. Indeed, our estimator provides a low false-positive rate when implemented on simulated data, and we will add such benchmarks to our revised appendix.
>
> **Other Concerns**
> The reviewer has also provided many useful suggestions about how to improve the clarity of the exposition and notation as well as identifying several places where additional details about the algorithm could be added. We thank the reviewer for their careful attention to detail while reviewing the article and look forward to implementing these changes.

---

### Official Review · Reviewer_CbdZ · 2021-07-17

**Rating:** 6
**Confidence:** 3

**Summary:**

This paper proposes a single-number measure called the maximum causal set effect to provide a simple and informative answer to complex causal queries that feature many causal variables. The measure is roughly the biggest difference alternative treatments can make, among treatments that appear sufficiently often in the population. An estimator for this quantity is developed and some useful statistical properties of the estimator are established. Empirical results are also presented, including an interesting application to real data from political science.

**Limitations And Societal Impact:**

Yes.

**Main Review:**

This is a solid paper, with original and useful results that are relevant to addressing an important question. One slightly misleading aspect is that the paper sometimes (e.g., in the abstract) describes the maximum causal set effect to be "the maximum difference in causal effects associated with two sets of causal variables of a researcher specified size." However, as I understand it, the two sets are not sets of variables, but rather sets of values of variables or sets of states, and the researcher specified size does not refer to the cardinality of a set but its probability. If my understanding is correct, I suggest such remarks should be reformulated. It seems to me that the need for such a measure as MCSE arises when there are a large number of alternative treatments. This will be the case if there are a large number of variables one can simultaneously control, but it may also be the case if some causal variable has many possible values.

Another puzzling point is on p. 4, lines 9-10, where a formula is introduced to quantify the effect by moderators or mediators. For moderators the formula makes sense to me, but I don't see why it is a sensible choice for mediators. Since mediators are affected by t_0, it is unclear to me what kind of effect the formula is quantifying.

The experiments take the PCA estimator as a baseline, but it does not seem fair to evaluate the PCA estimator against the MCSE, as the former is intended to capture a different aspect of a complex causal mechanism. I wonder if a single sample estimator of the maximum, despite its upward bias, will make a better comparison.

Minor comment: on p. 6, the top line has a typo: the last term should be about the Min set.


**Time Spent Reviewing:**

3

---

> ### Author Response · Authors · 2021-08-05
> **Response to Reviewer CbdZ**
>
> We thank the reviewer for their generous and thoughtful feedback. The reviewer makes several straightforward suggestions for how we can improve the article, and we look forward to implementing these changes.

---

> > ### Comment · Reviewer_CbdZ · 2021-09-03
> > **The case of mediators**
> >
> > Thanks for the response. However, after discussions with fellow reviewers, I am now worried that the case of mediators may not be as straightforward to accommodate as the response seems to indicate. I would appreciate an elaboration on how the authors plan to frame the mediators case. I decreased my score to 6 for now.

---

> > > ### Author Response · Authors · 2021-09-09
> > > **Many Mediators**
> > >
> > > Thank you for pushing us to give more thought to the many mediators case, which we now appreciate is more nuanced than we had initially believed. At this point, we're thinking that the paper would be most improved by removing the many mediators example and focusing on the many causes and moderators ones instead.
> > >
> > > What we had hoped to convey with the initial notation for $\tau\left(\mathcal{T}', \mathcal{T}''\right)$ in the mediators case was something like the average of a set of path specific effects (see this working paper for an example https://scholar.harvard.edu/files/xzhou/files/zhou-yamamoto_paths.pdf). Basically, each mediator $t_1 \dots t_k$ would identify a different causal path, so that $\tau\left(\mathcal{T}', \mathcal{T}''\right)$  would be defined as the difference in the average causal path effects associated with two different sets of mediators, $\mathcal{T}'$ and $\mathcal{T}''$. The MCSE would then provide a summary for the amount of dispersion in the path specific effects.
> > >
> > > There are two major issues with this approach. First, as you noted, the proper notation for defining a path specific effect is more complex than what we had used is the initial manuscript. The most common approach uses nested potential outcomes, which would require introducing a significant of amount of new notation. Second, the identification of many distinct path specific effects rests on the assumption that all of the mediators operate independently from each other, which is quite strong and doesn't seem likely to hold in many applications. So even though the setting of many mediators is a common one -- cases where there are many mediators believed to operate independently are likely quite rare. Our sense now is that the many mediators case is complex enough that it would be difficult to do it justice in the space available, and that the paper would be strengthened by focusing on the many causes and many moderators settings which are comparatively straightforward and more likely to be used by applied researchers anyways.
> > >
> > > That being said, if you or the other reviewers feel the many mediators case is an important one to explore, it would certainly be possible to reformulate the paper with potential outcomes notation and to provide some discussion about the needed identifying assumptions. A compromise might also be to mention the many mediators case in the conclusion while emphasizing the generality of the framework.

---

### Decision · Program_Chairs · 2021-09-27

**Decision:**

Accept (Poster)

**Comment:**

The paper considers a relevant problem, but reviewers have identified some sloppiness regarding causal claims:

The discussion of the mediators case on p. 4 struck several reviewers as confusing (and possibly confused.) The formula given between lines 119 and 120 is sensible with respect to the moderators case, but hardly makes sense with respect to the mediators case. The authors should either explain how to read or understand the formula in the mediators case so that it evidently expresses a meaningful effect differential, or use a different, clearly applicable formula for the mediators case, or at least restraint from suggesting that the said formula is also applicable to the mediators case.

Unfortunately, the authors replied to this issue in a very non-explicit way although it's a crucial point from the causal perspective.
We decided to accept this paper despite this problem. It should be emphasised, however, that we considered it serious enough to discuss rejection since a paper on causality should be careful about causal claims. We therefore ask the authors to consider the issue carefully in the revised version.